# Pre- and Post-Immigration Correlates of Alcohol Misuse among Young Adult Recent Latino Immigrants: An Ecodevelopmental Approach

**DOI:** 10.3390/ijerph16224391

**Published:** 2019-11-10

**Authors:** Eli Levitt, Bar Ainuz, Austin Pourmoussa, Juan Acuna, Mario De La Rosa, Juan Zevallos, Weize Wang, Pura Rodriguez, Grettel Castro, Mariana Sanchez

**Affiliations:** 1Department of Medical and Population Health Sciences Research, Herbert Wertheim College of Medicine, Florida International University, Miami, FL 33199, USA; elevi020@med.fiu.edu (E.L.); apour007@med.fiu.edu (A.P.); jacuna@fiu.edu (J.A.); juzevall@fiu.edu (J.Z.); rodrigup@fiu.edu (P.R.);; 2Center for Research on U.S. Latino HIV/AIDS and Drug Abuse (CRUSADA), School of Social Work, Robert Stempel College of Public Health and Social Work, Florida International University, Miami, FL 33199, USA; delarosa@fiu.edu (M.D.L.R.); wewang@fiu.edu (W.W.); 3Department of Health Promotion and Disease Prevention, Robert Stempel College of Public Health and Social Work, Florida International University, Miami, FL 33199, USA

**Keywords:** minority health, health disparities, social determinants of health, alcohol dependence, risk and protective factors, health

## Abstract

Latinos in the United States experience numerous alcohol-related health disparities. There is accumulating evidence that pre-immigration factors are associated with post-immigration alcohol use, but the explanation for health disparities remains unclear. The present study is a secondary analysis of data from the Recent Latino Immigrant Study (RLIS), the first community-based cohort study to examine the pre- to post-immigration alcohol use trajectories of young adult Latino immigrants during their initial years in the United States. Exploratory analysis and hierarchical multiple logistic regression were performed to assess associations between various pre- and post-immigration factors and alcohol misuse among young adult Latino immigrants early in the immigration process. Using an ecodevelopmental approach, we examined potential social and environmental determinants across multiple levels of influence associated with post-immigration alcohol misuse in this population. The study sample consisted of 474 young adult Latino immigrants between the ages of 18–34. The sample was comprised of the following national/regional origins: Cuban (43%), South American (28.7%), and Central American (28.3%). Approximately half of the sample (49.6%) reported a family history of substance use problems (FHSUP+). Participants who reported FHSUP+ and who engaged in alcohol misuse prior to immigrating to the US were more likely to engage in post-immigration alcohol misuse. Results revealed various social and environmental factors associated with pre-immigration alcohol misuse in this population. Study findings can inform culturally tailored prevention interventions aimed at mitigating problem drinking behaviors among young adult recent Latino immigrants.

## 1. Introduction

### 1.1. Alcohol Misuse among Latino Immigrants in the United States

Alcohol misuse is a significant cause of social problems, morbidity, and mortality in the United States (US) [1]. In 2016, alcohol consumption was responsible for 34,865 deaths in the United States [2]. The economic burden of excessive alcohol use was estimated to cost the United States $249 billion in 2010, with a subsequent increase of approximately 11% over four years [3]. Alcohol misuse is associated with several comorbidities, including liver cirrhosis, alcoholic hepatitis, alcohol-induced pancreatitis, fetal alcohol syndrome, dependence, and withdrawal [1].

In 2018, the population of adults who identified as Hispanic or Latino was estimated to be 59 million (18.1%) of the total US population. The Latino population in the US was expected to increase to 119 million (28.6%) by 2060 [4]. There is well-documented evidence regarding differences in drinking patterns across racial and ethnic groups; however, only limited explanations exist for the complex disparities [5].

Latinos in the US are less likely to consume alcohol compared to non-Latino Whites. However, Latinos who do drink are more likely to consume higher volumes of alcohol than non-Latino Whites [6,7]. As such, there is a need to examine the risk and protective factors that are antecedent to and perpetuate alcohol misuse that can inform the development of culturally-relevant interventions aimed at mitigating alcohol misuse among Latino immigrants.

A growing number of epidemiological studies have found that alcohol misuse among Latino immigrants tends to be substantially lower than alcohol misuse among their US-born counterparts [8,9]. Numerous investigations have observed increased rates of alcohol use among Latino immigrants as their time in the US increased [10]. The factors contributing to the development of alcohol misuse in recent Latino immigrants are not well understood. Previous studies have focused on post-immigration drinking factors influencing Latinos (predominantly of Mexican origin) [8]. This study adds analysis of drinking patterns among Latino immigrants from other national/regional origins, such as Cuba and South and Central America. These Latino immigrant subgroups have experienced remarkable growth in the US between 2000 and 2016 [11]. As such, this research aims to contribute to our understanding of the risk and protective factors associated with alcohol misuse among diverse Latino immigrant subgroups in the United States.

Examining risk and protective factors of alcohol misuse among young adult Latino immigrants is a particularly pressing public health issue. Young adulthood is a developmental period marked by increased responsibility and independence (e.g., career, marriage) [12]. These transitions may be particularly impactful for recent Latino immigrants forced to cope with the stressors involved in adapting to a new host society. The limited research conducted on recent Latino immigrant substance use has focused primarily on adolescents [13]. This avenue of research is particularly relevant to public health when considering that unlike non-Latino Whites, a population whose drinking peaks around ages 19–22, heavy drinking among Latinos peaks later and persists longer into adulthood [14].

The research literature has increasingly focused attention on the need to account for pre-immigration influences when examining alcohol use of Latino immigrants [15,16,17]. Immigrants arriving in the US as adults have often spent the bulk of their formative years in environments that are quite different from those of the US. By failing to consider the implications of such experiences, previous studies have disregarded the influence of the contexts in which many immigrants have lived before arrival that may affect their health behaviors after immigration. Accounting for the lived experienced of immigrants, both before and after immigration, allows for a more comprehensive approach that looks beyond immediate post-immigration factors to account for factors such as reasons for immigration and neighborhood context in the country of origin, as well as post-migration factors (e.g., acculturative stress and levels of social support) [9]. Using this heuristic will allow for a comprehensive approach to providing a more precise understanding of Latino immigrants’ experiences and the links between these experiences and alcohol misuse [18].

### 1.2. Research Aims

The present study aims to (1) examine the association between pre-immigration family history of substance use problems and post-immigration alcohol misuse among young adult recent Latino immigrants and (2) to assess other social and environmental factors associated with post-immigration alcohol misuse. Specifically, we expect that the pre-immigration family history of substance use problems will be associated with post-immigration alcohol misuse among young adult Latino immigrants. We also hypothesize that various social and environmental risk and protective factors before and after immigration will be associated with alcohol misuse use post-immigration.

### 1.3. Theoretical Framework

The conceptual framework for the present study is informed by ecodevelopmental models, based on Bronfenbrenner’s social-ecology theory of human development [19]. This theory posits that an individual’s behaviors are influenced by the settings in which they are embedded—their families, neighborhoods, nations—and that we cannot fully understand these behaviors without accounting for the additive effects of these contextual factors [20].

Social-ecological theory proposes that multiple factors influence an individual’s development across the following nested set of systems, which are listed from furthest to closest level of influence: macrosystems (the context related to the broader social environment, such as cultural, political, and economic influences), exosystems (the indirect environment in which the person does not participate directly but that impacts the functioning of important members of the individual’s life), mesosystems (contexts composed of interactions or connections, such as personal relationships), and microsystems (context related to the smallest and most immediate environment in which the person lives, such as the family).

Specifically, ecodevelopmental theory seeks to understand the multilevel factors predisposing and protecting youth from engaging in risky behaviors, including alcohol and drug use. The framework asserts that the risk and protective factors involved in substance use behaviors are influenced by interrelated processes within the microsystem, mesosystem, exosystem, and macrosystem [21]. As such, ecodevelopmental models focus on how multiple social contexts, and the interrelations between them, impact risk and resilience for the development of substance use [22]. In the present study, we focus specifically on the factors within the micro-, meso-, and macro-systems that may influence alcohol misuse among young adult recent Latino immigrants.

### 1.4. Social-Ecological Factors and Alcohol Misuse among Latino Immigrants

#### 1.4.1. Microsystemic Factors

According to ecodevelopment theory, the most immediate factors that place Latino immigrants at greater risk for alcohol misuse occur at the microsystemic level. This includes direct individual and family-level influences [21]. While family substance use history appears to be a rational initial step toward understanding the etiological pathways in the development of post-immigration alcohol use disorders, investigations in this line of research remain surprisingly scarce [23]. A study conducted by Blackson and colleagues found that biological parents’ histories of substance use problems were associated with higher scores on the Alcohol Use Disorders Identification Test (AUDIT) among recent Latino immigrants—both before and after immigration [23]. Another study revealed that alcohol dependence symptoms among US Latinos were associated with a family history of alcoholism. There is supportive evidence that heritability-liability estimates for alcohol dependence range between 48%–66% [18,24]. Researchers have posited that in the context of Latino immigrants, heritability-liability may not fully explain the etiology of alcohol misuse, although it may foreshadow the development of post-immigration alcohol use and misuse. The relationship between genetic predisposition and alcohol misuse with social and environmental factors has been inadequately characterized [23]. As such, there remains a need for future investigation examining social and environmental factors influencing the associations between genetics and ecodevelopmental factors in predicting alcohol misuse among Latino immigrants.

Other individual-level factors that have been found to influence substance abuse among Latinos include low socioeconomic status, undocumented immigration status, and marital status [25,26,27]. As with other racial and ethnic groups, demographic differences such as age and gender also predict greater substance abuse among Latino immigrants, with a higher prevalence of alcohol misuse among males and those of younger age [28] 

#### 1.4.2. Mesosystemic Factors

Ecodevelopmental theory posits that at the mesosystemic level, an individual’s behaviors are influenced by interpersonal or social relationships with others. The interpersonal relationships also encompass mutual relationships and the resulting interactions between components of the microsystem. Prominent theories of health behavior suggest that higher levels of social support may be beneficial in managing stress more effectively, in turn reducing the likelihood of using maladaptive coping strategies such as substance use [29]. Social support may be a particularly relevant determinant of health behavior among immigrants because these individuals are adjusting to a new receiving culture. Indeed, social support has been found to be an important mitigating factor in buffering alcohol use risk behaviors such as drinking and driving among Latino immigrants. It may be that investment in social relations facilitates the flow of information, as well the provision of tangible and intangible support, which, in turn, can make the immigration process less complicated and stressful, thereby reducing the likelihood of engaging in health-compromising behaviors such alcohol misuse {Sanchez, 2016 #21;Kissinger, 2013 #55}.

#### 1.4.3. Macrosystemic Factors

A further level of influence on an individual’s behavior is within the macrosystem, which relates to the broader socio-political, cultural, and economic system to which the individual belongs. Indeed, theorists have postulated that post-immigration alcohol misuse among immigrants is both directly and indirectly associated with the pre-immigration context, including the (a) social, economic, and political environment of the country of origin; (b) context of departure; and (c) reason for immigration [9]. In addition, post-immigration cultural stressors—including the social, economic, and political environment of the receiving society/community and experiences of discrimination and other acculturative stressors—can impact an immigrant’s susceptibility to alcohol misuse [9,30].

Specifically, acculturative stress has been linked to an array of negative health outcomes, including anxiety, depression, and alcohol abuse [31,32]. Acculturative stress consists of stressors experienced by individuals due to an incongruence of beliefs, values, and other cultural norms between their country of origin and country of reception [20]. This form of stress is usually triggered by factors such as language barriers; difficulties assimilating to beliefs, values, and norms of the host country; and perceived discrimination [33]. Undocumented immigration status and socioeconomic issues have been cited as the main sources of acculturative stress among Latino immigrants [34].

Cano and colleagues examined the relationship between acculturative stress and alcohol use severity among recent Latino immigrants. After controlling for demographic variables, significant positive associations were found between acculturative stress and alcohol use severity [35]. Moderating effects by gender were also found, whereby higher levels of acculturative stress were associated with alcohol use severity among men, but not among women [35]. Additionally, results from a recent national study found that Latino immigrants who experienced discrimination were more likely to report alcohol-related problem behaviors [16].

Although previous studies have documented the impact of risk and protective factors on alcohol use in Latino immigrants, crucial questions remain concerning the relationship between pre-immigration family history of substance use problems, post-immigration alcohol misuse, and the potential role that various social-ecological factors play in that association.

## 2. Materials and Methods

### 2.1. Study Design

The present study is a secondary data analysis from the first longitudinal prospective cohort study to examine sociocultural determinants of pre- to post-immigration alcohol use trajectories among young adult recent Latino immigrants in the US. Procedures were approved by the Social and Behavioral Institutional Review Board of a large public university in South Florida, and all participants provided written informed consent. Recruitment occurred between 2008 and 2010. Inclusion criteria were: (a) being between the ages of 18 and 34 years old; (b) self-identifying as Latino/a; (c) having immigrated to the US from a Latin American country within one year prior to interview; (d) residing within Miami-Dade County; and (e) intending to stay in the US for at least 3 years.

At baseline, *retrospective pre-immigration* data (T1) was collected in a sample of (n = 474) young adult recent Latino immigrants. A follow-up interview 24-months later collected *post-immigration* data (T2) (i.e., current socio-demographics, alcohol use patterns in the past 90 days, levels of acculturative stress, social support). T1 data was collected between 2008–2010, and the T2 assessment was completed between 2010–2012.

Respondent-driven sampling (RDS) was the primary recruitment strategy [36]. Due to the nature of the hard-to-reach population, including undocumented immigrants, RDS is considered to be the most appropriate method of recruitment [36]. Each participant (the seed) was asked to refer three individuals in his/her social network who met eligibility criteria. Seeds were recruited via flyers posted in Miami-Dade County neighborhoods with substantial Latino populations, during Latino health fairs in Miami Dade County, and through Latino Community Health Centers. This procedure was followed for a maximum of three legs per seed [36].

Data was collected through face-to-face survey administration. Trained bilingual research staff conducted the surveys. All surveys were conducted in Spanish and completed at a confidential, safe location agreed upon by both the interviewer and participant. Each survey required approximately 1 hour to complete. Interviews were audio recorded and reviewed by research assistants for quality control purposes. Participant compensation for their time was: $50 and $60 for T1 and T2, respectively [36]. This compensation was considered as non-coercive and not providing undue influence on study participants by the university’s research team and Institutional Review Board [36,37,38,39].

### 2.2. Measures

The present study uses T1 and post-immigration T2 data to account for pre- and post-immigration factors. As displayed in Table 1 the T1 variables included in the study were: alcohol misuse in the 90 days prior to immigration, pre-immigration family history of substance use problems, and neighborhood crime level in the participant’s country of origin. The remaining variables used in the present study are from T2 data, including the main study outcome: post-immigration alcohol misuse.

#### 2.2.1. Microsystem Measures

**Sociodemographics.** A demographics form assessed, in part, participants’ gender, language proficiency of Spanish and English, education level (1 = less than high school, 2 = high school, 3 = some training/college after high school, 4 = bachelor’s degree, 5 = graduate/professional studies), and average annual income of the last 12 months. The demographics form was completed at T1. Current documentation status was measured by 10 possible categories (e.g., temporary or permanent resident, temporary work visa, undocumented or expired visa) and then recoded into dichotomous variables: documented or undocumented.

**Family history of substance use problems (FHSUP).** Participants were surveyed on several aspects of family history of substance use problems at T1. The effect of substance use on the family members was assessed by determining health, law, or work problems related to the substance use in the father, mother, grandfather, and grandmother. The number of parents and grandparents with problems due to substance abuse was also determined. The sum of all family history of substance use items was measured by determining the number of all family members with substance use problems. These categories were recoded into a dichotomous variable: FHSUP+ = one or more family members with a history of substance use problems, or FHSUP− = no family history of substance use problems.

**Alcohol misuse**. Alcohol misuse in the past 90 days was assessed by the Alcohol Use Disorder Identification Test (AUDIT) score at both T1 and T2. The AUDIT is a 10 question screening tool developed by the World Health Organization (WHO) [40]. Sample items include, “How often do you have 5 or more drinks on one occasion?”; and “How often during the last year have you found that you were not able to stop drinking once you had started?” The AUDIT has been validated across gender, age, and culture. The AUDIT includes questions about recent alcohol use, alcohol dependence symptoms, and alcohol-related problems. Each item was rated on a five-point scale (0 to 4), totaling a score ranging from 0 to 40. According to the widely used AUDIT guidelines for use in primary health care scores ≥8 are indicative of alcohol misuse. This scoring has been shown to have a high level of sensitivity and specificity across cultures [40]. As such, in the present study, we measure alcohol misuse with a binary variable that was recorded as follows: sum score of 0–7 = (0) no alcohol misuse or a score of 8 or higher = (1) alcohol misuse.

#### 2.2.2. Mesosystem Measures

**Social support.** The Medical Outcome Social Support Survey (MOS [41]) was used to measure social support. The survey has been widely used to measure the effects of social support on various health outcomes [42,43,44]. The survey assesses four dimensions of social support, including emotional/informational, tangible, affectionate, and positive social interaction. Sample items include, “How often do you have someone you can count on to listen to you when you need to talk?”; and “How often do you have someone to do something enjoyable with?” The instrument contains 19 items set on a 5-point Likert-type scale (1 = *none of the time* to 5 = *all of the time*) with higher scores indicating more social support. Collected at T2. In the present study, the overall social support index score was calculated by averaging the scores for all 18 items included in the four subscales, and the score for the one additional item.

**Immigrated to the US alone or with family/friends.** At T1, participants were asked to respond to the following item: “Did you come to the United States alone, with your family or/and with friends?” These categories were recoded into a dichotomous variable: alone or with family or with friends.

#### 2.2.3. Macrosystem Measures

**Acculturative Stress**. At T2, acculturative stress [20,31,32,33,34] was measured by the validated Spanish version of the immigration stress subscale of the Hispanic Stress Inventory Scale-Immigrant Version. This scale has been widely used in Latino populations; it measures the stressful event experiences specific to immigrants. Sample items include “I have felt unaccepted by others due to my Latino culture”; and “Because my poor English, it has been difficult for me to deal with day to day situations.” The scale uses a five-point Likert-type scale format, with a subscale containing 18 questions. Questions are posed about whether or not the participant experienced a particular stressor, followed by a question regarding the appraisal of how stressful that event was (1 *= not at all* to 5 = *extremely*). The sum of the immigration stress frequency and immigration stress appraisal score was used to measure overall acculturative stress.

**Reason for immigration**. At T1, participants were asked, “Please rank in order of importance the reasons you immigrated to the United States with ‘1′ being the most important and ‘3′ being the least.” Response options were as follows: 1 = economic, 2 = to be with my family, 3 = political, 4 = other reason. These categories were recoded into a dichotomous variable: economic as the primary reason or non-economic as the primary reason.

**Pre-immigration neighborhood crime level.** Collected at T1. To assess exposure to neighborhood crime in their country of origin, participants were asked: “In the neighborhood in which you grew up or lived prior to coming to the US, how would you characterize the crime level? Response options were 1 = low, 2 = medium, and 3 = high.

### 2.3. Statistical Analysis

The characteristics of the exposed (FHSUP+) and non-exposed (FHSUP−) participants were documented to assess comparability. Descriptive analyses were conducted, including means and standard deviations for continuous variables and percentages for categorical variables. Two sample t-tests for continuous variables by alcohol misuse, and Chi-Square tests between alcohol misuse and each categorical variable, were conducted to assess the association between alcohol misuse and the variables. We then performed hierarchical multiple logistic regression (HMLR) to measure the associations of the predictor variables with alcohol misuse. Items were entered into the HMLR model in a specified order to compare the goodness of fit of the nested models. Predictor variables were grouped and entered into the HMR model in the following block order: (1) Microsystem variables, (2) Mesosystem Measures, and (3) Macrosystem variables. Log-likelihood ratio test statistics were calculated for model comparisons. A two-sided *p*-value < 0.05 was considered statistically significant. Statistical analyses were performed using the Statistical software SAS Enterprise Guide 7.1 (SAS Institute Inc., Cary, NC, USA).

### 2.4. Ethical Aspects for Human Subject Research

The project used a de-identified dataset that was previously collected for the purpose of research. The parent study was conducted in accordance with the Declaration of Helsinki, and the protocol was approved by the Institutional Review Board of a large public university in South Florida. Trained and supervised bilingual research staff obtained written consent from all participants and conducted all interviews in Spanish. All interviews were confidential and were completed at a location agreed upon by both the research staff and each participant.

## 3. Results

The study sample consisted of 474 participants (Mean age = 26.95, SD = 4.98; 52% female). The mean time since immigrating to the United States was 6.8 (SD = 3.2) months at baseline. Table 2 displays distributions of the sample characteristics by family history of substance use problems (FHSUP+/−) status. At the microsystem level: education, birth country/region, documentation status, and pre-immigration alcohol misuse differed by FHSUP status. With regard to mesosystemic variables, there were significant differences in social support and immigrating alone vs. with others by FHSUP status. At the macrosystems level, the primary reason for immigration, pre-immigration neighborhood crime level, and acculturative stress significantly differed by FHSUP status. Specifically, 55.3% of participants reporting FHSUP+ had education levels of a high school diploma or less. South and Central Americans had a significantly higher proportion of FHSUP+. Compared to those with FHSUP−, participants with FHSUP+ status more often reported alcohol misuse prior to immigration, having undocumented immigration status, immigrated to the US alone, immigrated for economic reasons, and reported living in neighborhoods with high levels of crime in their country of origin. Additionally, acculturative stress was significantly higher among those with FHSUP+ status, while social support was significantly lower. No significant associations were found between FHSUP and the following covariates: age, gender, marriage status, English proficiency, employment, or income in the past three months.

Table 3 presents the distribution of covariates by alcohol misuse. At the microsystems level, a chi-square test indicated a statistically significantly higher proportion of individuals with FHSUP+ status compared to FHSUP− status. The proportion of males was higher than females. Nearly half of the participants who had pre-immigration alcohol use also had alcohol misuse after immigration. At the mesosystem level, social support was significantly lower in people who had alcohol misuse after immigration than those who did not misuse alcohol. At the macrosystem level, alcohol misuse was greater among individuals immigrating for economic reasons than for non-economic reasons. No significant differences were found between alcohol misuse across age, marriage status, English proficiency, education, employment status, birth country/region, documentation status, pre-immigration neighborhood crime level, immigrating to the US alone, income, and acculturative stress.

Table 4 represents the adjusted odds ratios from the HMLR model. Results indicate that variables of the microsystem, including sociodemographics, pre-immigration alcohol misuse, and FHSUP, were significantly added to the regression model (Model 1, Δ-2 log-likelihood = 99.1, Δ degree of freedom = 12, *p* < 0.001), compared to the null (intercept-only) model. However, the second block, which contains the measures of mesosystem, did not significantly improve the observed model fit (Model 2, Δ-2 log-likelihood = 3.5, Δ degree of freedom = 2, *p* = 0.174). Furthermore, there was no significant difference in the model fit after adding the third block with the measures of macrosystem into the model (Model 3, Δ-2 log-likelihood = 4.7, Δ degree of freedom = 6, *p* = 0.583). From the estimated adjusted odds ratios, no significant association was found between alcohol misuse and family history of substance use problems (*p* > 0.05), adjusting for the covariates in the model. Results show that being a male (aORs = 1.97 and 1.93, *p* < 0.05) and having pre-immigration alcohol misuse (aORs ranged from 7.29 to 7.41, *p* < 0.001) was significantly associated with higher odds of alcohol misuse after immigration, adjusting for other variables in the model. No significant association was found between alcohol misuse with age, marital status, education, employment status, income, birth country, or documentation status (*p* > 0.05). In addition, alcohol misuse was not significantly associated with mesosystem measures, including social support and came to the US alone, or macrosystem measures that were acculturation stress, reason for immigration, English proficiency, and pre-immigration neighborhood crime level (*p* > 0.05).

## 4. Discussion

This study contributes to the knowledge of how pre- and post-immigration ecological factors contribute to alcohol misuse among Latino young adults early in the immigration process. Study findings indicate that at the most immediate microsystem level, young adults with a history of alcohol misuse were more likely to report alcohol misuse after immigrating to the United States. A particularly innovative aspect of this investigation is its capacity to assess pre-immigration factors. Pre-immigration findings offer a fuller contextual understanding of the lives of Latino young adults and can provide important background information that can be utilized to better meet the needs of the growing Latino immigrant population.

Latino immigrants arrive with a wide array of risks and resources (obtained through pre-immigration experiences). The few existing studies that have examined Latino immigrants’ alcohol use patterns prior to immigration have found that experiences before immigration considerably influence immigrants’ adaptation patterns to the US, including their alcohol use behaviors [45,46]. Results from the present study add to this emerging literature to underscore the need to account for pre-immigration personal and family history of substance use when assessing Latino immigrants’ health risk behaviors, including alcohol use [15,28,47]. Previous research suggests that family history poses a significant influence on other family members, as parental drinking may affect the drinking motives of youth and resultantly influence their alcohol use patterns [48,49,50,51]. This association may be particularly relevant among Latino immigrants. While the importance of family is found in many cultures, close family ties are a hallmark of Latino culture [52]. As such, family-level factors may have a stronger effect in predicting risk behaviors among Latinos. Future studies are needed to examine how various aspects of family functioning, such as family conflict or family cohesion, might moderate associations between FHSUP and alcohol misuse among Latino immigrants. These findings can help to inform the development of culturally relevant family-centered prevention interventions among Latino immigrants at risk for alcohol misuse.

Another individual-level factor associated with post-immigration alcohol misuse included gender. Specifically, the effects of gender found in the present study support previous evidence that Latino males are more likely to engage in alcohol misuse compared to females [6]. In a study of recent Latino immigrants conducted by Cano and colleagues (2018), gender was found to moderate the association between family cohesion and alcohol use, with males having a higher alcohol use severity [48]. The results from the present study build on this evidence and support the notion that recent Latino immigrant males with a family history of substance use problems may be at particular risk for engaging in alcohol misuse post-immigration. Recent Latino immigrants with FHSUP+ status were more likely to have undocumented immigration have lower levels of education compared to those without a family history of substance use problems.

Study findings also revealed the country/region of origin as another individual-level factor in which distinct differences were found. Specifically, in the present sample, Cuban immigrants were least likely to report FHSUP+ status, while immigrants from Central America had the highest proportions of participants reporting a family history of substance use problems. Despite widespread recognition of the cultural differences that exist between distinct Latino national groups, previous research on alcohol use among Latino immigrants has been conducted predominantly with Mexican immigrant samples [53,54,55]. Far less is known about the alcohol use trajectories of Cuban and South and Central American immigrants in the US From a public health perspective, this is particularly important because while Mexicans remain the largest US Latino immigrant group, shifts in immigration patterns within the past decade have indicated steep increases in immigrants from Central and South America arriving in the US. These studies should seek to provide a comprehensive understanding of the ecological factors, including structural and sociocultural influences that distinctly confer risk or protection in the development of alcohol misuse across Latino subgroups.

At the mesosystem level, social support was found to be significantly associated with both the pre-immigration family history of substance use problems and post-immigration alcohol misuse. Specifically, recent Latino immigrants reporting lower levels of social support were more likely to report FHSUP+ and post-immigration alcohol misuse. Results from the present study support previous evidence suggesting the higher levels of social support among recent Latino immigrants can assist in ameliorating alcohol-use related risk behaviors such as drinking and driving in this population [30]. These findings support the notion that future programs seeking to prevent alcohol misuse among Latino immigrants should incorporate strategies targeting interpersonal level factors aimed at increasing social support, including civic participation and reciprocity among residents. Programs aimed at supporting recent Latino immigrants in their initial settlement phase should focus on expanding support resources by actively connecting new immigrants with local neighborhood supports.

Within the broader social, cultural, political, and economic context at the macrosystem level, recent Latino immigrants with FHSUP+ status were more likely to report economic reasons for immigration, reside in neighborhoods with higher crime levels prior to immigration, and experience greater levels of post-immigration acculturative stress compared to those without a family history of substance use problems. Moving to the US for economic reasons was also associated with a greater likelihood of alcohol misuse.

Previous research has found that the acculturation process can also lead to significant changes in the Latino family structure over time (i.e., differential levels of acculturation between foreign-born parents and their US-born children) that place Latino immigrants at higher risk for substance use and other health-compromising behaviors [49]. Thus, understanding the role of immigration in Latino families—and the sequelae of these changes that can occur at the macrosystem level—is an important step in predicting and preventing alcohol misuse among Latino immigrants [56].

Specifically, acculturation has been vastly cited in the literature to account for changes in alcohol use behaviors of Latinos [1,5,17,35,48,50,57,58,59,60,61,62,63]. Bivariate analysis revealed a significant difference in the acculturative stress when comparing individuals with FHSUP+ compared to FHSUP−. However, no significant association between acculturative stress and alcohol misuse were found. Acculturation intends to capture the changes that occur in a group or individual who are assimilating into a new culture. A recent meta-analysis conducted by Lui and colleagues examined associations between acculturation and alcohol use among Latinos. Findings indicated that links between acculturation and alcohol tend to be more robust among women and within certain contexts [10]. Contextual factors that affect the phenomenon of acculturation and acculturative stress should be considered. One factor to take into account when interpreting results from the present study is the context of the immigrant-receiving community. Miami has a highly bicultural environment that is equally supportive of US and Latino cultural practices and families [64]. As such, acculturative stress may be less prominent in well-established immigrant-receiving communities like Miami that house strong ethnic enclaves. There remains a need for future research that examines how FHSUP interacts with sociocultural factors to impact alcohol misuse among Latino immigrants residing in new immigrant-receiving communities that may have limited tangible and intangible resources for recent immigrant populations [65].

The sociopolitical context of the immigrant-receiving community should also be considered. Our study sample was largely of Cuban origin. The Cuban Adjustment Act of 1966 allowed for any individual who fled Cuba and entered the US to pursue residency one year later. Subsequently, the ‘‘wet foot, dry foot’’ law that passed during the Clinton administration in 1995 offered Cuban immigrants a clear path to citizenship upon entering the US. Given the rights afforded to Cuban immigrants but not to other Latino immigrants, certain immigration stressors related to undocumented immigration status may not be relevant. It should be noted that data collected for the present study was completed prior to recent initiations in attempting to normalize diplomatic relations between the US and Cuba and the repeal of the “wet foot, dry foot” policy. It remains to be seen how these changes will impact immigration-related stressors among recent Cuban immigrants and its effects on their alcohol misuse.

### Limitations

The results of this study must be interpreted in the context of their limitations. First, respondent driven sampling was used to collect the Recent Latino Immigrant Study information, which does not ensure a representative sample in the study. However, this was the best approach to access and recruit study participants in the relatively hidden target population of immigrants, particularly those with undocumented immigration status. Second, data for family history of substance use problems and pre-immigration alcohol misuse was collected retrospectively, making this information susceptible to recall bias. Attaining a prospective cohort for the information of interest, such as a family history of substance use was not plausible; thus, a retrospective collection of the data was deemed the best method. A loss to follow-up is a common limitation in retrospective studies. However, due to the excellent retention rate in our sample, selection bias is not likely to have significantly impacted the study.

Despite its limitations, this study sheds light on the association between pre- and post-immigration correlation of alcohol use patterns among recent Latino immigrants in the United States. The early stages of acclimating to a new country often involve challenges and stresses that test coping skills. As such, this transition period may offer a key point in time for prevention-oriented screening and intervention for immigrants with a history of alcohol-use related problems.

## 5. Conclusions

Alcohol misuse is a substantial public health concern. The present study uniquely documents how an individual’s family history and pre-immigration experiences, as well as ecological factors across various systems of influence, impact post-immigration alcohol misuse. By including the family history of Cuban and South and Central American immigrants, this study addresses a fundamental gap in the literature regarding these understudied Latino subgroups.

Alcohol misuse can be identified routinely in general practice and emergency care. People are often asked about alcohol consumption during new patient registrations, general health checks, and other health screening procedures. Information about an individual’s family history may contribute to the primary and secondary prevention of alcohol misuse in individuals who may have fewer supports and fewer financial resources to make healthy decisions. There is a need to research and develop culturally relevant strategies for identifying recent young adult Latino immigrants with a family history of substance use problems early in the immigration process to assist health providers with early detection to prevent alcohol misuse for this at-risk population. Such strategies could take the form of incorporating preventive screening of family history of substance use problems into public health programs that service recent Latino immigrants.

We anticipate that findings from this study will advance knowledge regarding social and environmental factors that are antecedent to and perpetuate alcohol misuse among recent Latino immigrants. Resources such as brief screenings for FHSUP and alcohol misuse among recent Latino immigrants can help identify those who would benefit from early interventions such as motivational interviewing, alcoholics anonymous, and follow-up with a primary care physician. These resources can help mitigate the economic and health consequences of alcohol misuse. Additional culturally-informed research on the social and environmental factors associated with alcohol misuse in this population is of high public health significance, as it may inform prediction, prevention, and treatment of problem-drinking behaviors among the largest ethnic minority group in the US.

## Figures and Tables

**Table 1 ijerph-16-04391-t001:** Description of study measures.

		Sample Item/Description	Categories
**Microsystem**			
Age	T1	How old are you?	Recoded binary as ≥25 or ≤25
Gender	T1	What is your gender?	Binary: (0) Male or (1) Female
Marital Status	T2	What is your current marital status? 5 categories (i.e., married, single, divorce, widowed)	Recoded as binary (0) Single or (1) Married/Partnered
Education	T2	What is the highest grade you completed in school?	5-point Likert scale ranging from (1) Less than high school to (5) Graduate/Professional Studies
Employment	T2	Are you currently employed?	Binary: (1) yes or (0) no
Income	T2	Total income in past 3 months	In total US dollars
Birth country	T1	Where were you born? (List of 17 possible Latin American countries provided)	Recoded as (1) Cuban, (2) South American or (3) Central American
Documentation status	T2	Which of the following best describes your current immigrant status? (List of 10 options (i.e., permanent resident, temporary resident, work visa, students visa, undocumented)	Recoded as binary (0) Undocumented or (1) Documented
Alcohol misuse	T1	AUDIT (Alcohol Use Disorders Identification Test). 10 items, Likert Scale. Sample item: In the past 12 months “How often do you have 5 or more drinks on one occasion?” Sum score recoded to cut-off of ≥ 8 indicating alcohol misuse	Recoded binary as ≥8 = (1) alcohol misuse or ≤8 = (0) no alcohol misuse
Family history of substance use problems	T1	Effect of substance use on the family members was assessed by determining health, law, or work problems related to the substance use in the father, mother, grandfather, and grandmother.	FHSUP+ = one or more family members with a history of substance use problems. FHSUP− = no family history of substance use problems.
**Mesosystem**			
Medical Outcomes Study Social Support Scale	T2	19 items on a 5-point Likert-type scale (1 = *none of the time* to 5 = *all of the time*) Sample item: “How often do you have someone you can count on to listen to you when you need to talk?”	Sum of all items calculated for total scale score.
Came to the U.S. alone	T1	“Did you come to the United States alone, with your family or/and with friends?”	(0) Alone (1) With family or with friends
**Macrosystem**			
Acculturative stress	T2	Hispanic Stress Index-Immigration Stress Subscale: 18 items ask whether a stressor occurred (1) yes or (0) no. Follow up appraisal item rates how stressful the experience was on a 5-point Likert-type scale ranging from (1 *= not at all* to 5 = *extremely*). Sample item: “Because my poor English, it has been difficult for me to deal with day to day situations.”	Sum of frequency score and appraisal score.
Reason for immigration	T1	“Please rank in order of importance the reasons you immigrated to the United States ‘1 being the most important and ‘3′ being the least.”	Recoded as (1) Economic versus (0) non-economic as primary reason for immigration
English proficiency	T2	How well do you speak/understand English?	4-point Likert scale (1) Understand/speak a little to (4) Understand/speak very well
Pre-immigration neighborhood crime level	T1	“In the neighborhood in which you grew up or lived prior to coming to the US, how would you characterize the crime level?	3-point Likert scale (1) Low (2) Medium (3) High

**Table 2 ijerph-16-04391-t002:** Characteristics of recent Latino immigrants with or without family history of substance use problems (FHSUP), Miami.

	Family History of Substance Use Problems (FHSUP)		
Variables	^a^ FHSUP−	^a^ FHSUP+	*p*-Value
	(n = 239)	(n = 235)	
	%	%	
Age (years)			0.237
≤25	37.2	42.6	
>25	62.8	57.4	
Gender			0.262
Female	48.5	43.4	
Male	51.5	56.6	
Marital status			0.737
Married/partnered	37.7	36.2	
Single	62.3	63.8	
English proficiency			0.065
Speak/Understand a little	27.8	33.6	
Understand fairly well	48.5	37.9	
Speak English well or very well	23.6	28.5	
Education			0.021
Less than high school or high school diploma	43.1	55.3	
Some training/college after high school	39.7	28.9	
Bachelor’s degree or advanced education	17.2	15.7	
Employment			0.198
No	15.5	20.0	
Yes	84.5	80.0	
Birth country/region			<0.001
Cuba	57.6	28.1	
South America	26.7	30.7	
Central America	15.7	41.1	
Immigration reason			0.003
Economic reasons	51.9	65.5	
Not economic reasons	48.1	34.5	
Documentation status			<0.001
Documented	84.1	60.0	
Undocumented	15.9	40.0	
Pre-immigration neighborhood crime level			<0.001
Low	52.7	34.9	
Medium	34.7	32.3	
High	12.6	32.8	
Immigrated to the US alone			0.017
Yes	43.1	54.0	
No	56.9	46.0	
Pre-immigration alcohol misuse			<0.001
Yes	18.8	43.0	
No	81.2	57.0	
	Mean (^b^ SD)	Mean (^b^ SD)	*p*-value
Income (total in past 3 months)	7294 (5117)	8011 (6289)	0.174
Acculturative stress	5.2 (3.9)	6.9 (5.0)	<0.001
Social support scale total score	4.2 (0.9)	3.9 (1.0)	<0.001

*^a^* FHSUP, family history of substance use problems; ^b^ SD, standard deviation; Participants were grouped by exposure; the percentages are reported by column.

**Table 3 ijerph-16-04391-t003:** Characteristics of recent Latino immigrants with or without alcohol misuse.

	Alcohol Misuse				
Variables	No (n = 376)	%	Yes (n = 98)	%	*p*-Value
Family history of substance use problems					0.010
No	201	84.1	38	15.9	
Yes	175	74.5	60	25.5	
Age (years)					0.656
≤25	148	78.3	41	21.7	
>25	228	80.0	57	20.0	
Gender					<0.001
Female	195	89.4	23	10.6	
Male	181	70.7	75	29.3	
Marital status					0.055
Married/partnered	147	84.0	28	16.0	
Single	229	76.6	70	23.4	
English proficiency					0.956
Speak/Understand a little	114	78.6	31	21.4	
Understand fairly well	163	79.9	41	20.1	
Speak English well or very well	98	79.7	25	20.3	
Education					0.088
High school diploma or less	182	78.1	51	21.9	
Some training/college after high school	125	76.7	38	23.3	
Bachelor’s degree or greater	69	88.5	9	11.5	
Employment					0.281
No	63	75.0	21	25.0	
Yes	313	80.3	77	19.7	
Birth country/region					0.160
Cuba	167	83.1	34	16.9	
South America	103	76.9	31	23.1	
Central America	99	75.0	33	25.0	
Immigration reason					0.015
Economic reasons	210	75.5	68	24.5	
Not economic reasons	166	84.7	30	15.3	
Documentation status					0.348
Documented	275	80.4	67	19.6	
Undocumented	101	76.5	31	23.5	
Pre-immigration neighborhood crime level					0.227
Low	172	82.7	36	17.3	
Medium	124	78.0	35	22.0	
High	80	74.8	27	25.2	
Immigrated to the US alone					0.091
Yes	175	76.1	55	23.9	
No	201	82.4	43	17.0	
Pre-immigration alcohol misuse					<0.001
Yes	79	54.1	67	45.9	
No	297	90.5	31	9.5	
	Mean	SD ^a^	Mean	SD	*p*-value
Income (total in past 3 months)	7522.1	5725.9	8137.8	5718.8	0.344
Acculturative Stress	5.9	4.4	6.3	5.0	0.463
Social Support Scale Total Score	4.1	0.9	3.8	1.0	0.009

^a^ SD, standard deviation.

**Table 4 ijerph-16-04391-t004:** Hierarchical multiple logistic regression model for predicting alcohol misuse (N = 467).

	^a^aOR (95% ^b^CI)		
Variable	Model 1	Model 2	Model 3
	Microsystem	Mesosystem	Macrosystem
Block 1: Microsystem			
Age: >25 years old vs. ≤25 years old	0.92 (0.53,1.58)	0.91 (0.53,1.57)	0.87 (0.50,1.50)
Gender: Male vs. Female	1.97 (1.07,3.62) *	1.93 (1.03,3.59) *	1.85 (0.98,3.49)
Marital Status: Single vs. Married/partnered	1.72 (0.96,3.06)	1.68 (0.94,3.01)	1.74 (0.96,3.13)
Education:			
^a^ Some training after H.S. vs H.S. or <	1.55 (0.83,2.93)	1.77 (0.92,3.4)	1.89 (0.96,3.71)
Bachelor’s degree or > vs. H.S. or <	0.44 (0.17,1.11)	0.48 (0.19,1.25)	0.53 (0.20,1.44)
Employment: Yes vs. No	0.64 (0.33,1.23)	0.59 (0.30,1.16)	0.60 (0.30,1.21)
Income (total in past 3 months)	1.00 (1.00,1.00)	1.00 (1.00,1.00)	1.00 (1.00,1.00)
Birth country:			
South America vs. Cuba	1.25 (0.63,2.49)	1.09 (0.54,2.2)	1.01 (0.48,2.13)
Central America vs. Cuba	1.1 (0.45,2.68)	0.99 (0.40,2.43)	0.94 (0.37,2.38)
Documented vs. Undocumented status	1.44 (0.65,3.18)	1.53 (0.68,3.41)	1.75 (0.75,4.1)
Pre-immigration alcohol misuse: Yes vs. No	7.41 (4.14,13.27) **	7.29 (4.06,13.08) **	7.35 (4.05,13.34) **
^b^ FHSUP+ vs. FHSUP−	1.17 (0.67,2.06)	1.16 (0.66,2.03)	1.15 (0.65,2.05)
Block 2: Mesosystem			
Social support scale total score		0.78 (0.58,1.05)	0.77 (0.57,1.03)
Came to the U.S. alone		1.21 (0.70,2.07)	1.22 (0.71,2.11)
Block 3: Macrosystem			
Acculturative stress			1.02 (0.96,1.08)
Reason for immigration: Economic vs. To be with my family			1.51 (0.83,2.74)
English proficiency:			
Understand fairly well vs. Speak/understand a little			1.18 (0.62,2.25)
Speak English well or very well vs. Speak/understand a little			1.09 (0.49,2.42)
Pre-immigration neighborhood crime level:			
Medium vs. Low			1.08 (0.58,2.01)
High vs. Low			1.06 (0.52,2.16)

^a^ aOR = adjusted odds ratio, ^b^ CI = confidence interval, * *p* < 05, ** *p* ≤ 001. -2 Log Likelihood = 380.8 for Model 1, Δ-2 Log Likelihood = 99.1 and Δ degree of freedom = 12 for Model 1 compared to the null model, *p* < 001. Δ-2 Log Likelihood = 3.5 and Δ degree of freedom = 2 for Model 2 compared to Model 1, *p* = 174. Δ-2 Log Likelihood = 4.7 and Δ degree of freedom = 6 for Model 3 compared to Model 2, *p* = 583. ^a^ H.S. = High school. ^b^ FHSUP = Family History of Substance Use Problems.

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
