# Peer review of "Pre- and Post-Immigration Correlates of Alcohol Misuse among Young Adult Recent Latino Immigrants: An Ecodevelopmental Approach"

_ijerph, 2019, doi:10.3390/ijerph16224391_

Round 1

Reviewer 1 Report

I would like to thank authors for an interesting presentation of their research results. In current form, the following parts of the paragraph are described with care and attention to detail. This research, amongst other, raises the problem of alcohol misuse by immigrants which, as the authors also mention, is not usually analyzed in this group. The added value of the work is placing the whole analysis design concept in the theoretical base which is the theory of ecological systems. Thanks to this procedure, the reader can obtain information about the direct impact of subsequent elements of the system on the development of alcohol misuse abut also about relations between subsequent systems which results in this behavior.

To summarize, the strength of the study is that the analysis design based on scientific theory. The authors summarized the discussion by listing some limitations of their study, what shows on their high awareness and distance in interpreting results. The manuscript flows well, and in most of the paper, the writing style is clear.

Aside of the advantages, some minor changes should be made to the manuscript:

Verse 146 - in my opinion, it should be added that relationships also refer to mutual relationships and the resulting interactions between components of the microsystem. Verse 185, the phrase "2.1 study design" should be moved to the next verse The paragraph regarding the type of data collected in subsequent phases of the study (verse 197 from the "The present study combines ... to the end of the text in verse 202) should be moved to the beginning of section 2.2 If possible (depending on the restrictions on the number of tables in the text) please create a table with the measures listed. Currently, part of the text is quite illegible. At first glance, one cannot see which data comes from the T1 tests and which from the T2 tests. Additionally, placing data in the Table would reduce the volume of this fragment of text, thus improving the readability of the paper. If the authors do not decide to amend it, please add the test round from which the variable originates (T1 or T2 or both) for each variable described

Tab. EXAMPLE (please note that this is only an example of such table)

Type of the system*

Measure (with shortcut)

Examples of items OR content of question OR description of question

Categories (if applicable)

Reference (if applicable)

…………………. T1

Microsystem

Mesosystem

Macrosystem

…………………. T2

Microsystem

Mesosystem

Macrosystem

*according to the theory of ….

Acronyms of two series of studies from which the data were analyzed should be cited earlier than in 196 verse. I propose to add these acronyms as follows: - verse 193, after the expression "retrospective pre-immigration data" - here in brackets (T1); - in verse 194 after "post-immigration data" in brackets (T2). Next, authors can only use acronyms without quoting the name of the study stage. Verse 314, it seems to me that there should be a semicolon instead of a comma, in brackets after the word female Verse 315 - do we have an exact data about central tendency with SD? In my opinion the description of the results should be precise. The authors should write exact results (mean, sd..) instead of presenting an average situation. Or if this variable was categorical it should be written that the largest percentage were people who immigrated to US XX year ago. Verse 324 instead of "more likely to report" should be "more often reported alcohol misuse etc .. In addition, I suggest you remove the percentages from brackets because they are a repetition of the data from the table. this applies to verses from 314 to 341. Please pay attention to text editing, e.g. missing spaces (verse 168 - parenthesis with reference number) or double spaces (verse 323 before “compared…”), verse 365 “knowledge” instead of “skowledge”. Please read the text once again carefully.

Author Response

Comment

Author’s Response

Verse 146 - in my opinion, it should be added that relationships also refer to mutual relationships and the resulting interactions between components of the microsystem.

The following statement has been incorporated into the revised manuscript: “The interpersonal relationships also encompass mutual relationships and the resulting interactions between components of the microsystem.”

Verse 185, the phrase "2.1 study design" should be moved to the next verse.

 The paragraph regarding the type of data collected in subsequent phases of the study (verse 197 from the "The present study combines ... to the end of the text in verse 202) should be moved to the beginning of section 2.2

We appreciate the reviewer’s attention to detail. Based on the reviewers feedback the phrase 2.1 study design was moved to the subsequent verse.

The paragraph regarding the data used in the study has also been moved to section 2.2 measures.

If possible (depending on the restrictions on the number of tables in the text) please create a table with the measures listed.

As per the reviewer's suggestions,the revised manuscript now includes a measures table (Table 1).

Measures: Currently, part of the text is quite illegible. At first glance, one cannot see which data comes from the T1 tests and which from the T2 tests. Additionally, placing data in the Table would reduce the volume of this fragment of text, thus improving the readability of the paper. If the authors do not decide to amend it, please add the test round from which the variable originates (T1 or T2 or both) for each variable described

We thank the reviewer for their helpful feedback. The revised manuscript now clarifies the time points in which the study variables were assessed. We also include a table with measures description in the revised manuscript. 

Acronyms of two series of studies from which the data were analyzed should be cited earlier than in 196 verse. I propose to add these acronyms as follows: - verse 193, after the expression "retrospective pre-immigration data" - here in brackets (T1); - in verse 194 after "post-immigration data" in brackets (T2). Next, authors can only use acronyms without quoting the name of the study stage.

The revised manuscript now includes the T1 and T2 as acronyms as per the reviewer’s suggestion.  

Verse 314, it seems to me that there should be a semicolon instead of a comma, in brackets after the word female

This edit has been incorporated in the revised manuscript and can now be found in verse 313.

Verse 315 - do we have an exact data about central tendency with SD? In my opinion the description of the results should be precise. The authors should write exact results (mean, sd..) instead of presenting an average situation. Or if this variable was categorical it should be written that the largest percentage were people who immigrated to US XX year ago.

The mean time with SD since immigration to the US at baseline has been included in the revised manuscript.  

Verse 324 instead of "more likely to report" should be "more often reported alcohol misuse etc .. In addition, I suggest you remove the percentages from brackets because they are a repetition of the data from the table. this applies to verses from 314 to 341.

As per the reviewer’s request verse 324 has been revised as follows: “more often reported” and the brackets with percentages have been removed from the updated manuscript.

Please pay attention to text editing, e.g. missing spaces (verse 168 - parenthesis with reference number) or double spaces (verse 323 before “compared…”), verse 365 “knowledge” instead of “skowledge”. Please read the text once again carefully.

These oversights in editing have been corrected in the revised manuscript.  

Reviewer 2 Report

The definition of recent latino immigrants wasn’t provided. The citation for the longitudinal study that that provided the secondary data was not provided. Methods is well explained with clear variables and measurement criteria. The sampling procedure was however, not well explained. The consent procedure was not well explained. The payment of money 50 – 60 USD to study participants is not clear, was it approved in the IRB protocol? No explanation/justification provided for using a cut of 8 for alcohol misuse. How was the cut of 8 determined, and why was it used? What was the prevalence of alcohol misuse? The justification for using odds ratios instead of prevalence ratios should be used. It would have been appropriate to use prevalence ratios instead of odds ratios. Table 3 is not very meaningful. It would have been appropriate for the authors to present a Table with both the bivariable and multivariable analysis, and the proportions of those that misused and those that didn’t misuse alcohol Table 2 doesn’t make sense especially with the p – values. The p values do not mean anything in that table without odds ratios. The 2 can be revised to be bivariable analysis of factors associated with alcohol misuse. Unadjusted odds ratios or prevalence ratios.

Author Response

Comment

Author’s Response

The definition of recent Latino immigrants wasn’t provided

A definition for Latino has been provided in the revised manuscript.  

The citation for the longitudinal study that that provided the secondary data was not provided. Methods is well explained with clear variables and measurement criteria.

We thank the reviewer for catching this oversight. The citation for the parent study has been included in the revised manuscript.

The sampling procedure was however, not well explained. The consent procedure was not well explained. The payment of money 50 – 60 USD to study participants is not clear, was it approved in the IRB protocol?

We thank the reviewer for their valuable comments and have including additional information regarding the study procedures in the revised manuscript.  

No explanation/justification provided for using a cut of 8 for alcohol misuse. How was the cut of 8 determined, and why was it used?

A justification for the cut off 8 has been added (see verse 265).  Additionally, a citation to the AUDIT guidelines has been included in the revised manuscript.

What was the prevalence of alcohol misuse?

Alcohol misuse was found in 31.1% (n=147) of the participants before immigration and 20.7% (n=98) of the participants at time three.

The justification for using odds ratios instead of prevalence ratios should be used. It would have been appropriate to use prevalence ratios instead of odds ratios.

The odds ratios were directly estimated from logistic regression. We chose to use logistic regression for data analysis because of: (1) the outcome (alcohol misuse) is binary with yes/no responses (2) then event rate was not rare (>10%). In order to obtain prevalence ratios, other regression methods need to be applied, such as Poisson or log-binomial models. However, the models do not fit our data well due to the distribution of the outcome.

Table 3 is not very meaningful. It would have been appropriate for the authors to present a Table with both the bivariable and multivariable analysis, and the proportions of those that misused and those that didn’t misuse alcohol Table 2 doesn’t make sense especially with the p – values. The p values do not mean anything in that table without odds ratios. The 2 can be revised to be bivariable analysis of factors associated with alcohol misuse. Unadjusted odds ratios or prevalence ratios.

While we appreciate and have taken the reviewer’s important comments into consideration, Table 3 was included in response to comments from reviewer’s in the previous round of reviews. As such, we have respectfully decided to keep Table 3 in the revised manuscript.